# Fermented Beverage Benefits: A Comprehensive Review and Comparison of Kombucha and Kefir Microbiome

**DOI:** 10.3390/microorganisms11051344

**Published:** 2023-05-19

**Authors:** Ann Qi Chong, Siew Wen Lau, Nyuk Ling Chin, Rosnita A. Talib, Roseliza Kadir Basha

**Affiliations:** Department of Process and Food Engineering, Faculty of Engineering, Universiti Putra Malaysia (UPM), Serdang 43400, Selangor, Malaysia; angieannqi@gmail.com (A.Q.C.); raclsw96@gmail.com (S.W.L.); rosnita@upm.edu.my (R.A.T.); roseliza@upm.edu.my (R.K.B.)

**Keywords:** kombucha, kefir, fermentation, micro-organisms, benefits, health

## Abstract

Beverage fermentation is an ancient ritual that has been practised for millennia. It was slowly disappearing from households and communities due to the advancement of manufacturing technology and the marketing of soft drinks until the recent revival of the beverage fermentation culture due to an increase in the demand for health drinks amid the COVID-19 pandemic. Kombucha and kefir are two well-known fermented beverages that are renowned for their myriad of health benefits. The starter materials for making these beverages contain micro-organisms that act like microscopic factories producing beneficial nutrients that have antimicrobial and anticancer effects. The materials modulate the gut microbiota and promote positive effects on the gastrointestinal tract. Due to wide variations in the substrates and types of micro-organisms involved in the production of both kombucha and kefir, this paper compiles a compendium of the micro-organisms present and highlights their nutritional roles.

## 1. Introduction

Fermented beverages have a long history of being enjoyed by diverse cultures worldwide, offering a wide variety of options ranging from boza [1], cider [1], and kvass [2] to mead [3] and sake [4]. Kombucha and kefir are notable additions to this category, contributing to the rich tapestry of fermented beverages that are cherished by people across the globe. Both these cultured drinks have been derived from ancient rituals and passed down since the dawn of time by people of many different cultures and backgrounds. Kombucha is believed to have originated in northeastern China, in the Manchuria region [5,6]. Long notable in the East, kombucha has been consumed for over 2000 years and is said to have travelled to Europe and the rest of the world via trade paths [7,8]. Like kombucha, kefir has been around for umpteen years, dating back to the early people of the Caucasus Mountains [9]. Kefir grains were passed down from generation to generation among Caucasus tribes and were considered as a source of family wealth [9,10]. Around 1960s, kefir made its way to the western world [10]. With the expansion of commercial routes, both beverages gained prominence around the world. While both beverages were distributed to various countries, many names of kombucha and kefir were discovered. Kombucha is also recognised as haipao, teakwass, tea fungus, Manchurian mushroom, and kambotscha [8,11], whilst kefir is known as kephir, kefyr, kiaphur, kefer, kepi, knapon, and kippi in different places [12].

The two ancient fermented beverages became popular as they possess characteristics of ‘functional food’. The phrase, ‘functional food’ was coined in the early 1990s to describe food that not only provides essential nourishment but also contains biologically active compounds that can promote good health [13]. The development of functional foods has had a positive impact on the food industry, as more people have become health-conscious and are seeking foods that provide specific health benefits beyond basic nutrition [8]. Amid the COVID-19 pandemic, the important role of functional foods came to the forefront. In 2019, the global markets of kombucha [14] and kefir [15] were valued at USD 1.84 and 1.23 billion and are anticipated to grow to USD 10.45 and 1.84 billion by 2027, at a CAGR of 23.2% and 5.4%, respectively. The recent entry of the two largest beverage companies in the world, PepsiCo and Coca-Cola, into the kombucha market is a reflection of the growing demand for functional beverages [11]. Kefir, on the other hand, has gained a good reputation and is being dubbed the ‘yogurt of the 21st century’ [16,17]. The growth in popularity of kombucha and kefir is anticipated to persist due to these positive responses.

The increased awareness of less sugar consumption is further encouraging the market growth of kombucha and kefir. Consumers are becoming more wary of overly sweet formulations, creating an opportunity for sour and bitter flavour notes to gain prominence in the beverage industry. Both kombucha and kefir inject new flavours into the beverage sector with unique and distinct taste sensations, making them appealing to different people with varying taste preferences. Kombucha tastes tangy and is slightly sour with a hint of sweetness, like apple cider vinegar [18]. The carbonation and effervescence of kombucha give it a crisp, sparkling, and refreshing mouthfeel. Kefir, on the other hand, has a creamy, smooth, and thick texture with a slightly sour taste that is comparable to drinkable yoghurt [17,19]. Nevertheless, the flavour and texture of naturally fermented kombucha and kefir can vary depending on the substrate, fermentation duration, and starter cultures used.

The starter cultures consisting of a diverse array of living bacteria and yeast play a crucial role in the production of both kombucha and kefir. They operate like tiny biochemical factories, while transforming simple ingredients into healthy and revitalising beverages, and they also produce an abundance of organic acids, amino acids, polyphenols, antibiotic substances, vitamins, and enzymes. However, both the starter cultures contain different consortiums of micro-organisms that synthesise different metabolites, contributing to different health benefits. Hence, this review aims to provide insight into the fermentation of kombucha and kefir, comparing the production of both beverages and exploring the potential health benefits associated with the different micro-organisms involved. Gaining a better understanding of the roles of beneficial bacteria will help the development of these new ingredients in nutritionally optimised foods, which prioritise consumer health. This provides a significant opportunity for advancement in the field and highlights the importance of ongoing research in this area. The information about kombucha and kefir is presented in Figure 1, which provides a summary of these two products, while Table 1 outlines the common benefits of kombucha and kefir.

## 2. Production of Kombucha and Kefir

The production of both kombucha and kefir begins with a sugar-containing liquid and a culture of bacteria and yeast to kick off fermentation. Kombucha is fermented with the infusion of sweetened black tea followed by the incorporation of a starter culture, known as a ‘SCOBY’ (Symbiotic Culture of Bacteria and Yeast) [7,79,80] for a period of 7 to 10 days [80]. The SCOBY is a three-dimensional yellow-brown cellulosic zoogleal mat that contains acetic acid bacteria (AAB) and osmophilic yeast in a symbiotic relationship [80]. Appropriate quantities of ingredients are necessary to support the growth and activity of the SCOBY. Generally, 1.5–6 g/L of tea is added to 1 L of boiling water, which is then filtered and steeped with 70–90 g/L of sugar [81,82,83]. After the mixture is cooled, the SCOBY is added, along with some previously fermented kombucha beverage (10% *v*/*v*) [81,83] to lower the pH to below 4.6 [82]. The acidic environment is critical for preventing the growth of undesirable and contaminating microbes that could compromise the quality and safety of the final product [82,83]. The combination of these ingredients leads to a complex series of chemical and microbial reactions that drive the fermentation process [82] during which a new SCOBY layer forms on the surface of the liquid until it covers the solution completely [83]. The SCOBY layer becomes thicker subsequently [83], and when the uppermost layer of the SCOBY is fully formed, it can be harvested and used to initiate a new batch of kombucha [8,83]. The size of the SCOBY formed is influenced by both the surface area and depth of the culture medium [83].

The role of kefir grains in kefir is like what a SCOBY does for kombucha. Kefir grains are the primary ingredient required to make kefir. Kefir grains, unlike wheat, barley, or rice, are gelatinous polysaccharides containing a culture of bacteria and yeast known as kefiran. Kefir grains range in size from 0.3 to more than 2.0 cm in diameter and have irregular, folded, or uneven surfaces; their shape and colour (white to yellowish) resemble cauliflower florets [12,17]. They are also characterised by a tough and resilient texture composed of branched chains of glucose and galactose, as a result of the microbial metabolism of milk lactose with an acidic taste [84,85]. The ideal ratio between the grains and fermentation substrate of animal milk is between 1:30 and 1:50 (*w/v*) [86,87]. The grains are added to the milk and left to ferment for 18 to 24 h [88,89]. The fermentation process occurs typically within temperatures between 8 and 25 °C in a partially closed container [87]. Traditionally, fermentation has been performed in bags made with animal hides, which are regularly shaken to ensure that the milk and kefir grains are well mixed [12,90]. The micro-organisms responsible for the fermentation process mostly dwell on the surface of the grains, encompassing bacteria and yeast cells. The biomass of kefir grains slowly increases during the milk fermentation [12]. Once the fermentation process is completed, the kefir grains are separated from the liquid by filtration with a sieve and can be reused to make subsequent batches of kefir [87]. The properties of the grains will be passed on to the following generations of new grains [16].

Kombucha and kefir are both fermented beverages made with a starter material that can be reused to make additional batches. The cultures aid in transforming, harnessing, and accessing the vitality and nutrients in the substrates. Kombucha is primarily made with black tea [91], whereas kefir has traditionally been made from milk substrates from cows, ewes, goats, or other types of milk [92]. Black tea provides a SCOBY with the necessary compounds, particularly purine derivatives, such as caffeine and theophylline, amply present in the tea [93]. Nevertheless, kombucha has undergone a massive transformation and is available in various variations today [94]. Studies have presented satisfactory outcomes in terms of their biochemical properties and kinetics [5,6,95,96]. With strong and increasing interest in kombucha, it has been suggested that the kombucha SCOBY could be used as an atypical starter in dairy products [97,98,99]. The kombucha SCOBY as a potential starter source in milk fermentation yields products similar to kefir or yoghurt [97,99]. The microbial species from the kombucha SCOBY and kefir grains can easily adapt to different substrates and lead to the development of new products. As a result, for vegan, lactose-intolerant, and dairy-product-allergic consumers, nondairy substrates are made available for kefir fermentation [92]. Water kefir, an alternative to milk kefir, is cultivated in sugared water, with or without fruit extracts [92,100,101]. Fruit juices, vegetable juices, and alcohols are suitable media for the development of nondairy-fermented kefir. Table 2 summarises the different substrates available for the fermentation of kombucha and kefir. This information can serve as a starting point for ingredient exploration and development. As technology advances, new and improved methods for fermenting different substrates can lead to potential functional product innovations.

The main pathways of substrate conversion into numerous products for both beverages were identified and are summarised in Figure 2. In kombucha fermentation, sucrose from the medium is first hydrolysed to simple sugars, namely glucose and fructose, by the enzyme invertase (β-fructofuranosidase, EC 3.2.1.26), which is primarily produced by yeast species, such as *S. cerevisiae* [5,169,170]. Yeasts synthesise ethanol and carbon dioxide as metabolites from the resultant monosaccharides, which is then oxidised by AAB to produce acetic acid over the following days [171,172]. The actual processes in kombucha are catalysed by the two primary metabolites, ethanol and acetic acid; acetic acid (most characteristic product of kombucha) promotes yeast to make ethanol, whereas ethanol stimulates the growth and production of AAB [5,18,155]. Concurrently, AAB are responsible for cellulose synthesis from glucose and fructose, which makes up the SCOBY [5,91,150,172]. Additionally, d-glucose at the C6 position and the aldehyde group of the β-d-glucose at the C1 position are both enzymatically oxidised by AAB, resulting in significant amounts of glucuronic acid (GlcUA) and d-glucano-δ-lactone, respectively [169]. This latter metabolite is hydrolysed into gluconic acid by microbial enzymes [169]. Other organic acids, such as oxalic, succinic, malic, and citric acids, play important roles in the biological processes by acting as intermediates or end products in metabolic pathways [96,173]. In some circumstances, metabolically active lactic acid bacteria (LAB) can produce a significant amount of lactic acid [174]. Apart from the main metabolites, chemical constituents present in kombucha originate from the substrate itself, where their structures can be altered and transformed into new components during fermentation [5]. With tea substrate, kombucha contains most of the tea ingredients, such as various polyphenols, flavonols, catechins, catechin gallates, adenine, caffeine, theobromine, theophylline, gallic acids, tannins, gallotannin, potassium, manganese, fluoride ions, vitamins A, B, C, E, and K, and amino acids, particularly theanine [5,18]. Vitamin C, the most common vitamin found in kombucha beverages, is assumed to be derived from glucose and synthesised by bacteria [96]. The changing profiles during the fermentation process and the end products are complex. The compositions depend on many factors, including the raw materials utilised, carbon source, amount of tea used, microbial makeup of the SCOBY, and conditions of fermentation process (time, temperature, and pH) [169].

Kefir fermentation is a complex interplay of various microbial strains, substrate compounds, grain-to-substrate ratios, and environmental conditions, all of which can substantially affect the final properties of kefir beverages [175,176]. The ingredients used in making kefir are milk and kefir grains, which contain a mixture of yeast, LAB, and AAB [12,16,175]. During kefir fermentation, yeast is responsible for the lactose conversion to ethanol and carbon dioxide [9,176]. Lactose is also hydrolysed through the lactose permeate system (LPS) or the phosphoenolpyruvate-dependent phosphotransferase system (LPPDPS), present mainly in LAB [177,178]. The LPS hydrolyses lactose into glucose and galactose, while the LPPDPS hydrolyses lactose to glucose and galactose-6-phosphate, via the action of the β-galactosidase enzyme [177,178]. LAB utilise two primary pathways for lactose metabolism: the homofermentative pathway, resulting in the production of lactic acid, and the heterofermentative pathway, producing a combination of end products including carbon dioxide, ethanol, acetic acid, and lactic acid [177,178,179]. Acetic acid is also produced by AAB [180]. The accumulation of organic acids from these pathways allows kefir to delay the development of pathogenic micro-organisms considerably [85]. Beyond the formation of organic acids, LAB also produce flavour compounds, such as acetaldehyde, diacetyl, and acetoin [12]. At the same time, kefir fermentation also results in the production of other metabolites, including amino acids, peptides, and vitamins [181]. Although some vitamins, for instance, vitamin B_1_, vitamin B_12_, folic acid, vitamin K, and riboflavonoid, increase in concentration during fermentation, others are utilised by the microbiota [12].

In summary, kombucha and kefir are two fermented beverages that have similarity in terms of the incorporation of a sweetened medium and a starter culture containing yeast and bacteria to embark on the fermentation process. The sugar in the starting ingredients transforms into a plethora of metabolites including organic acids, alcohols, and gases. Despite the similarities in production, there are differences between kombucha and kefir in terms of the distinctive types of yeast and bacteria that result in variations in their nutritional profiles and flavours.

## 3. Microbial Communities

The exact microbiota of kombucha and kefir are not well defined as they depend on the sources of the starter culture, growth conditions, processing techniques, and types of substrates [17,85]. The origin of the starter culture plays one of the most crucial roles in determining the micro-organisms present in the fermented beverages as it can be influenced by the climatic and environmental conditions in a specific region that leads to the dominance of certain strains of micro-organisms and the evolution of distinct types of fermentations in that area. In addition, the type of substrate has a significant impact on the divergence of microbial communities as it provides sources of nutrients and energy for the micro-organisms to grow and multiply. Different substrates with varying levels of nutrients can influence the growth of various micro-organisms. Table 3 and Table 4 present the microbial composition in kombucha- and kefir-fermented products, respectively, showcasing the correlation with different substrates and locations. These tables offer a thorough understanding of the micro-organisms present in these fermented beverages, highlighting the impact of the substrates and locations on their composition.

The relationship between bacteria and yeast in both consortia of fermented beverages is complex. Both commensal and amensal associations may occur between them at the same time. The SCOBY in kombucha is a symbiotic association of AAB and yeast with dominant AAB genera of *Komagataeibacter*, *Acetobacter, Gluconacetobacter*, and *Gluconobacter* and yeast genera of *Schizosaccharomyces*, *Candida*, *Zygosaccharomyces*, *Saccharomyces,* and *Brettanomyces* [11,182]. The basic microbiota of kefir grains contain LAB, such as the genera *Lactococcus*, *Leuconostoc,* and *Streptococcus*, both the homofermentative and heterofermentative species of *Lactobacillus*, yeast from the genera of *Kluyveromyces* and *Saccharomyces,* and some AAB [84,85,175,176]. LAB represented 83–90% of the microbial count in kefir grains, while yeasts accounted for 10–17% [16,17]. Although LAB, such as *Lactobacillus* spp. and *Leuconostoc* spp., are also detected in kombucha occasionally [11], their abundance is not as dominant as in kefir grains. In short, kombucha is a more abundant source of AAB, while kefir is a richer source of LAB.

Beneficial micro-organisms contribute to the health benefits of kombucha and kefir, either directly as probiotics or indirectly when they release useful metabolites (biogenics) [50]. Probiotics can be defined as micro-organisms that are alive, nonpathogenic, and advantageous for the host in an appropriate dosage, according to the definitions of the Food and Agriculture Organization and World Health Organization [72]. Some examples of probiotics isolated from kombucha and kefir are bacteria, including *Lb. acidophilus*, *Lc. casei*, *Lc. rhamnosus*, *Bifidobacterium lactis*, and *Bacillus coagulans* and yeasts including *Km. marxianus*, *S. cerevisiae,* and *S. boulardii* [8,72,183]. Apart from that, metabolites produced by microbes during fermentation, including proteolytic enzymes, organic acids (e.g., glucuronic acid), and exopolysaccharides, further enhance health [28,37,184].

**Table 3 microorganisms-11-01344-t003:** Microbial composition in fermented kombucha produced using different substrates from various locations.

Substrate	Source of SCOBY	SCOBY (Starter Culture)			Fermented Kombucha		
		Bacteria	Yeast	References	Bacteria	Yeast	References
**Tea**							
Black tea	Australia		*Z* *. bailii*	[171]		*Z. bailii*	[171]
			*Sz. pombe*			*Sz. pombe*	
			*T* *. delbreuckii*			*T. delbreuckii*	
			*R* *. mucilaginosa*			*R. mucilaginosa*	
			*Brett. bruxellensis*			*Brett. bruxellensis*	
			*C. stellata*			*C. stellata*	
	Brazil	*Kb. hansenii*	*Z. bailii*	[148]	*Kb. hansenii*	*Z. bailii*	[148]
		*Kb. europaeus*	*R. mucilaginosa*		*Kb. europaeus*	*R. mucilaginosa*	
		*Kb. xylinus*	*S. cerevisiae*		*Kb. xylinus*	*S. cerevisiae*	
		*Sg. melonis*	*Malassezia* spp.		*Sg. melonis*	*Malassezia* spp.	
		*Kb. rhaeticus*			*Kb. rhaeticus*		
	Canada	*Gluconacetobacter* spp.	*Dekkera* spp.	[185]	*Acetobacter* spp.	*Dekkera* spp.	[185]
			*Zygosaccharomyces* spp.		*Gluconacetobacter* spp.	*Zygosaccharomyces* spp.	
			*Davidiella* spp.		*Lactobacillus* spp.		
			*Pichia* spp.		*Lactococcus* spp.		
			*Wallemia* spp.		*Leuconostoc* spp.		
			*Lachancea* spp.		*Bifidobacterium* spp.		
			*Leucosporidiella* spp.		*Thermus* spp.		
	China	*A. pasteurianus*	*Z. bailii*	[186]			
		*G. xylinus*					
	France	*O. oeni*	*D. bruxellensis*	[174]	*O. oeni*	*D. bruxellensis*	[174]
		*Lq. nagelii*	*D. anomala*		*Lq. satsumensis*	*D. anomala*	
		*A. okinawensis*	*H. valbyensis*		*G. eurapaeus*	*H. valbyensis*	
		*G. eurapaeus*	*C. boidinii*		*G. liquefaciens*	*Wm. anomalus*	
		*G. hansenii*	*S. uvarum*		*Gb. cerinus*	*Zt. florentina*	
		*G. intermedius*	*P. membranifaciens*		*Gb. oxydans*		
		*Gb. oxydans*	*Z. bailii*		*Gb. saccharivorans*		
		*A. tropicalis*	*Zt. florentina*		*Gb. oboediens*		
	Germany		*Brett*. *lambicus*	[187]			
			*Zygosaccharomyces* spp.				
			*Saccharomyces* spp.				
			*C. krusei*				
			*C. albicans*				
			*Sd. ludwigii*				
			*C. kefyr*				
		*Kb. hansenii*	*Z. lentus*	[188]			
	India	*A. aceti* MTCC 2945	*Z. bailii*	[189]			
			*Brett. claussenii*				
	Indonesia				*Komagataeibacter* spp. DS1MA.62A	*Brett. bruxellensis*	[190]
					*Kb. xylinus*		
					*Kb. saccharivorans*		
					*G. saccharivorans*		
	Ireland	*Acetobacter* spp.	*Zygosaccharomyces* spp.	[185]	*Acetobacter* spp.	*Dekkera* spp.	[185]
		*Gluconacetobacter* spp.	*Pichia* spp.		*Gluconacetobacter* spp.	*Zygosaccharomyces* spp.	
		*Lactobacillus* spp.	*Kazachstania* spp.		*Lactobacillus* spp.		
		*Lactococcus* spp.	*Kluyveromyces* spp.		*Lactococcus* spp.		
		*Thermus* spp.	*Naumovozyma* spp.		*Thermus* spp.		
		*Enterococcus* spp.	*Z. lentus*		*Propionibacterium* spp.		
		*Propionibacterium* spp.					
	Mexico	*Gb. oxydans*	*S. cerevisiae* ATCC 18824	[129]			
		*A. aceti*	*Km. marxianus* NRRL Y-8281				
			*Brett. bruxellensis* NRRL Y-1411				
	Saudi Arabia		*C. guilliermondii*	[191]		*C. guilliermondii*	[191]
			*C. kefyr*			*C. kefyr*	
			*C. krusei*			*C. krusei*	
			*Sd. ludwigi*			*Sd. ludwigi*	
			*C. colleculosa*			*C. colleculosa*	
	Singapore	*A. xylinum*	*Kloeckera* spp.	[147]			
		*A. xylinoides*	*Sz. pombe*				
		*A. aceti*	*S. ludwigii*				
		*A. pausterianus*	*S. cerevisiae*				
		*B. gluconicum*	*Torulaspora* spp.				
			*Z. bailii*				
			*Pichia* spp.				
	Sri Lanka	*Gluconoacetobacter* spp.	*Zygosaccharomyces* spp.	[192]			
		*Acetobacter* spp.	*Dekkera* spp.				
		*Lactobacillus* spp.	*Pichia* spp.				
		*Leuconostoc* spp.					
		*Lactococcus* spp.					
		*Bifidobacterium* spp.					
	United Kingdom	*Acetobacter* spp.	*D. bruxellensis*	[185]	*Gluconacetobacter* spp.		[185]
		*Gluconacetobacter* spp.	*D. anomala*		*Lactobacillus* spp.		
		*Lactococcus* spp.	*Kz. unispora*		*Lactococcus* spp.		
		*Lactobacillus* spp.			*Thermus* spp.		
		*Enterococcus* spp.					
		*Kb. intermedius*	*Z. parabailli*	[152]	*Kb. intermedius*		[152]
		*Komagataeibacter* spp.	*Brett. bruxellensis*		*Komagataeibacter* spp.		
	United States	*Gluconacetobacter* spp.		[185]	*Gluconacetobacter* spp.		[185]
					*Lactobacillus* spp.		
					*Lactococcus* spp.		
					*Allobaculum* spp.		
					*Ruminococcaceae Incertae Sedis*		
	Unknown	*Komagataeibacter* spp.	*Candida* spp.	[150]	*Komagataeibacter* spp.	*Candida* spp.	[150]
		*Gluconobacter* spp.	*Lachancea* spp.		*Gluconobacter* spp.	*Lachancea* spp.	
			*Kluyveromyces* spp.		*Lyngbya* spp.	*Kluyveromyces* spp.	
			*Debaryomyces* spp.		*Bifidobacterium* spp.	*Debaryomyces* spp.	
			*Pichia* spp.		*Collinsella* spp.	*Pichia* spp.	
			*Waitea* spp.		*Enterobacter* spp.	*Waitea* spp.	
			*Eremothecium* spp.		*Weissella* spp.	*Eremothecium* spp.	
			*Meyerozyma* spp.		*Lactobacillus* spp.	*Meyerozyma* spp.	
			*Zygowilliopsis* spp.			*Zygowilliopsis* spp.	
			*Saccharomyces* spp.			*Saccharomyces* spp.	
			*Saccharomycopsis* spp.			*Saccharomycopsis* spp.	
			*Hanseniaspora* spp.			*Hanseniaspora* spp.	
			*Kazachstania* spp.			*Kazachstania* spp.	
			*Starmera* spp.			*Starmera* spp.	
			*Merimbla* spp.			*Sporopachydermia* spp.	
			*Sporopachydermia* spp.			*C. stellimalicola*	
			*Sugiyamaella* spp.			*C. tropicalis*	
			*C. stellimalicola*			*C.parapsilosis*	
			*C. tropicalis*			*Lh. thermotolerans*	
			*C. parapsilosis*			*Lh. fermentati*	
			*L. thermotolerans*			*Km. marxianus*	
		*A. senegalensis*	*Brett. bruxellensis*	[193]	*A. senegalensis*	*Brett. bruxellensis*	[193]
		*A. tropicalis*	*Brett. anomalus*		*A. tropicalis*	*Brett. anomalus*	
		*A. musti*	*P. fermentans*		*A. musti*	*P. fermentans*	
		*A. peroxydans*	*C. sake*		*A. peroxydans*	*C. sake*	
		*Gb. oxydans*	*Lh. fermentati*		*Gb. oxydans*	*Lh. fermentati*	
		*G. europaeus*	*Sz. pombe*		*G. europaeus*	*Sz. pombe*	
		*Kb. xylinus*	*Z. bailii*		*Kb. xylinus*	*Z. bailii*	
		*Kb. rhaeticus*			*Kb. rhaeticus*		
		*Kb. intermedius*			*Kb. intermedius*		
		*Kb. saccharivorans*			*Kb. saccharivorans*		
		*O. oeni*			*O. oeni*		
		*Pseudomonas* spp.			*Pseudomonas* spp.		
Green tea	Brazil	*Kb. hansenii*	*Z. bailii*	[148]	*Kb. hansenii*	*Z. bailii*	[148]
		*Kb. europaeus*	*R. mucilaginosa*		*Kb. europaeus*	*R. mucilaginosa*	
		*Kb. xylinus*	*S. cerevisiae*		*Kb. xylinus*	*S. cerevisiae*	
		*Sg. melonis*	*Malassezia* spp.		*Sg. melonis*	*Malassezia* spp.	
		*Kb. rhaeticus*			*Kb. rhaeticus*		
		*Komagataeibacter* spp.	*Saccharomyces* spp.	[161]	*Komagataeibacter* spp.	*D. bruxellensis*	[161]
		*Acetobacter* spp.	*D. bruxellensis*		*Acetobacter* spp.	*Brettanomyces* spp.	
		*Gluconobacter* spp.	*Brettanomyces* spp.		*Gluconobacter* spp.	*Zygosaccharomyces* spp.	
		*Zm. mobilis*	*Zygosaccharomyces* spp.		*Zm. mobilis*	*H. guilliermondii*	
		*Liquorilactobacillus* spp.	*H. guilliermondii*		*Liquorilactobacillus* spp.		
		*Ligilactobacillus* spp.			*Ligilactobacillus* spp.		
	France	*O. oeni*	*D. anomala*	[174]	*O. oeni*	*C. boidinii*	[174]
		*Lq. nagelii*	*D. bruxellensis*		*Lq. nagelii*	*D. anomala*	
		*A. okinawensis*	*H. valbyensis*		*A. lovaniensis*	*D. bruxellensis*	
		*A. tropicalis*	*S. cerevisiae*		*A. peroxydans*	*H. valbyensis*	
		*G. eurapaeus*	*Z. bailii*		*A. syzygii*	*T. microellipsoides*	
		*G. intermedius*	*Zt. florentina*		*A. tropicalis*	*Zt. florentina*	
		*G. xylinus*			*G. xylinus*		
		*Gb. oxydans*			*Tc. sakaeratensis*		
					*Gb. oxydans*		
	United Kingdom				*Komagataeibacter* spp.		[152]
					*Gb. entanii*		
					*Kb. intermedius*		
Black tea	China	*Gluconacetobacter* spp.	*S. cerevisiae*	[23]			
Green tea		*Lp. plantarum*					
Tea powder							
Rooibos tea	United Kingdom				*Komagataeibacter* spp.		[152]
					*Gb. entanii*		
					*Kb. intermedius*		
					*Kb. rhaeticus*		
**Others**							
Cheese whey	Mexico	*G. xylinus*	*Brett. bruxelensis*	[129]			
			*Km. marxianus*				
			*S. cerevisiae*				

Abbreviations: A., *Acetobacter*; B., *Bacterium*; Brett., *Brettanomyces*; C., *Candida*; D., *Dekkera*; Gb., *Gluconobacter*; G., *Gluconacetobacter*; Kz., *Kazachstania*; Km., *Kluyveromyces*; Kb., *Komagataeibacter*; Lp., *Lactiplantibacillus*; Lh., *Lachancea*; Lq., *Liquorilactobacillus*; O., *Oenococcus*; P., *Pichia*; R., *Rhodotorula*; S., *Saccharomyces*; Sd., *Saccharomycodes*; Sz., *Schizosaccharomyces*; Sg., *Sphigomonas*; Tc., *Tanticharoemia*; T., *Torulospora*; Wm., *Wickerhamomyces*; Z., *Zygosaccharomyces*; Zt., *Zygotorulaspora*; Zm., *Zymomonas*.

**Table 4 microorganisms-11-01344-t004:** Microbial composition in fermented kefir produced using different substrates from various locations.

Substrate	Source of Grains	Kefir Grains (Starter Culture)			Fermented Kefir		
		Bacteria	Yeast	References	Bacteria	Yeast	References
**Milk**							
Camel	Turkey				*Lactobacillus* spp.		[139]
					*Lactococcus* spp.		
Cow	Canada	*L. lactis* subsp. *lactis*	*Km. marxianus* subsp. *fragilis*	[138]	*Lactococcus* spp.		[138]
		*L. lactis* subsp. *cremoris*			*Lactobacillus* spp.		
		*L. lactis* subsp. *lactis* bv. diacetylactis			*Enterococcus* spp.		
		*Leu. mesenteroides* subsp. *cremoris*					
		*Lp. plantarum*					
		*Lc. casei*					
	Spain	*Lv. brevis*	*T. delbrueckii*	[85]			
		*Weissella viridescens*	*S. cerevisiae*				
		*Lt. kefiri*	*Kz. unispora*				
		*Lc. paracasei* subsp. *tolerans*	*C. kefyr*				
		*Lc. rhamnosus*	*C. holmii*				
		*Lc. paracasei* subsp. *paracasei*	*C. friedrichii*				
		*Lm. fermentum*	*Km. lactis*				
		*Lb. acidophilus*	*P. fermentans*				
		*Lb. gasseri*					
		*L. lactis* subsp. *lactis*					
		*Leuconostoc* spp.					
		*Sc. thermophilus*					
	Turkey				*Lactobacillus* spp.		[139]
					*Lactococcus* spp.		
					*Lactobacillus* spp.		[140]
					*Lactococcus* spp.		
					*Lb. acidophilus*		
					*Bifidobacterium* spp.		
Goat	Turkey				*Lactobacillus* spp.		[140]
					*Lactococcus* spp.		
					*Lb. acidophilus*		
					*Bifidobacterium* spp.		
Rice	Poland	*L. lacti*		[120]			
		*Leuconostoc* spp.					
		*Sc. thermophilus*					
		*Lactobacillus* spp.					
Bovine	Poland				*Lactobacillus* spp.	*C. kefyr*	[146]
Caprine					*Lactococcus* spp.	*Kz. unispora*	
Ovine					*L. lactis* subsp. *lactis*		
					*L. lactis* subsp. *lactis* bv. diacetylactis		
					*L. lactis* subsp. *cremoris*		
					*Leu. mesenteroides* subsp. *cremoris*		
					*Lt. kefiri*		
Milk	Argentina	*Lp. plantarum*	*Saccharomyces* spp.	[194]			
(unknown)		*Lt. kefiri*	*Km. marxianus*				
		*L. lactis* subsp. *lactis*					
		*Leu. mesenteroides*					
		*Acetobacter* spp.					
		*Lt. parakefiri*					
		*L. lactis* subsp. *lactis* bv. diacetylactis					
			*S. cerevisiae*	[183]			
			*Kz. unispora*				
			*P. occidentalis*				
			*Km. marxianus*				
	Brazil				*Lt. kefiri*	*Km. lactis*	[135]
					*Lt. parabuchneri*	*Kz. aerobia*	
					*Lc. paracasei*	*Lh. meyersii*	
					*Lc. casei*	*S. cerevisiae*	
					*L. lactis*		
					*A. lovaniensis*		
	Bulgaria	*L. lactis* subsp. *lactis*	*Km. marxianus* subsp. *lactis*	[16]	*L. lactis* subsp. *lactis*	*Km. marxianus* subsp. *lactis*	[16]
		*Sc. thermophilus*	*S. cerevisiae*		*Sc. thermophilus*	*S. cerevisiae*	
		*Lb. delbrueckii* subsp. *bulgaricus*	*C. inconspicua*		*Lb. delbrueckii* subsp. *bulgaricus*	*C. inconspicua*	
		*Lb. helveticus*	*C. maris*		*Lb. helveticus*	*C. maris*	
		*Lc. paracasei* subsp. *paracasei*			*Lc. casei* subsp. *pseudoplantarum*		
		*Lv. brevis*			*Lv. brevis*		
	China	*Bacillus subtilis*	*Km. marxianus*	[195]			
		*L. lactis*	*S. cerevisiae*				
		*Lt. kefiri*	*P. kudriavzevii*				
		*Leu. lactis*	*Kz. unispora*				
		*Lp. plantarum*	*Kz. exigua*				
		*A. fabarum*	*M. guilliermondii*				
		*A. okinawensis*	*S. cerevisiae*	[184]			
		*Leu. pseudomesenteroides*	*Kz. unispora*				
		*L. lactis* subsp. *lactis*					
	Croatia		*Kz. unispora*	[196,197]		*Kz. unispora*	[196,197]
			*T. delbrueckii*			*T. delbrueckii*	
	Germany		*Km. marxianus*	[196,197]		*Km. marxianus*	[196,197]
			*S. turicensis*			*S. turicensis*	
	Iran	*Lv. brevis*	*C. kefyr*	[19]			
		*Lt. kefiri*	*S. lactis*				
		*Lc. casei*	*S. fragilis*				
		*Lp. plantarum*					
		*L. lactis*					
		*Leu. mesenteroides*					
	Ireland	*Lactococcus* spp.		[195]	*Lactococcus* spp.		[195]
		*Lb. kefiranofaciens*			*Lb. kefiranofaciens*		
		*Lt. kefiri*			*Lt. kefiri*		
		*Lt. parabuchneri*			*Lt. parabuchneri*		
		*Lb. kefiranofaciens* subsp. *kefirgranum*			*Lb. kefiranofaciens* subsp. *kefirgranum*		
		*Lb. helveticus*			*Lb. helveticus*		
		*Lb. acidophilus*			*Lb. acidophilus*		
		*Lt. parakefiri*			*Lt. parakefiri*		
	Poland		*Kz. unispora*	[196,197]		*Kz. unispora*	[196,197]
			*S. turicensis*			*S. turicensis*	
			*Brett. anomalus*			*Brett. anomalus*	
	Switzerland		*C. kefyr*	[196,197]		*C. kefyr*	[196,197]
			*S. turicensis*			*S. turicensis*	
			*C. colliculosa*			*C. colliculosa*	
**Others**							
Apple	United	*A. fabarum*		[114]	*Lq. satsumensis*		[114]
	Kingdom				*A. fabarum*		
					*A. suratthaniensis*		
Cocoa	Brazil	*Lb. kefiranofaciens* subsp. *kefirgranum*	*Km. marxianus*	[105]	*Lb. kefiranofaciens* subsp. *kefirgranum*	*Km. marxianus*	[105]
(*Theobrom*		*Lp. plantarum*	*S. cerevisiae*		*Lp. plantarum*	*S. cerevisiae*	
*cacao* L.)		*Lm. fermentum*	*Kz. unispora*		*Lm. fermentum*	*Kz. unispora*	
		*Lb. kefiranofaciens* subsp. *kefiranofaciens*			*Lb. kefiranofaciens* subsp. *kefiranofaciens*		
		*Acetobacter* spp.			*Acetobacter* spp.		
Fig	United	*Lt.* *hilgardii*		[114]	*A. persici*	*P. kudriavzevii*	[114]
	Kingdom	*Lq. satsumensis*					
		*Gb. cerinus*					
		*A. fabarum*					
		*A. syzgii*					
		*Kb. saccharivorans*					
Raisin	United	*Lq. satsumensis*	*P. kudriavzevii*	[114]	*A. orientalis*		[114]
	Kingdom	*Lq. oeni*			*A. syzygii*		
		*A. fabarum*			*Gb. cerinus*		
		*Gb. cerinus*					
		*Gb. oxydans*					
Sucrose	United	*Lq. satsumensis*		[114]	*Gb. cerinus*		[114]
	Kingdom	*A. syzgii*					
		*Gb. cerinus*					
	Malaysia	*Lt. hilgardii*		[198]			
		*Sb. harbinensis*					
		*A. lovaniensis*					
		*A. tropicalis*					
		*Lq. satsumensis*					
		*Lc. zeae*					
		*O. oeni*					
		*Gb. oxydans*					
		*Kb. hansenii*					
Sucrose	United Kingdom	*Gb. oxydans*	*P. membranifaciens*	[114]	*Lq. nagelii*	*P. membranifaciens*	[114]
(nitrogen		*Lq. nagelii*	*Zt. florentina*		*A. indonesiensis*	*S. cerevisiae*	
limitation)		*A. persici*			*A. orientalis*	*Zt. florentina*	
		*Lc. paracasei*			*A. syzygii*	*P. kudriavzevii*	
					*Gb. oxydans*		
					*A. persici*		
					*Lc. casei*		
					*Lc. paracasei*		
					*A. cerevisiae*		
					*A. fabarum*		
					*A. papayae*		
					*A. suratthaniensis*		
	Ireland	*Lq. nagelii*		[114]	*Lq. nagelii*	*P. membranifaciens*	[114]
					*A. fabarum*	*Zt. florentina*	
					*A. indonesiensis*		
					*A. orientalis*		
					*A. tropicalis*		

Abbreviations: A., *Acetobacter*; C., *Candida*; Gb., *Gluconobacter*; Kz., *Kazachstania*; Km., *Kluyveromyces*; Kb., *Komagataeibacter*; Lh., *Lachancea*; Lc., *Lacticaseibacillus*; Lp., *Lactiplantibacillus*; Lb., *Lactobacillus*; L., *Lactococcus*; Lt., *Lentilactobacillus*; Lv., *Levilactobacillus*; Leu., *Leuconostoc*; Lm., *Limosilactobacillus*; Lq., *Liquorilactobacillus*; M., *Meyerozyma*; P., *Pichia*; S., *Saccharomyces*; Sb., *Schleiferilactobacillus*; Sc., *Streptococcus*; T., *Torulospora*; Zt., *Zygotorulaspora*.

## 4. Health Benefits

### 4.1. Antioxidant

Excessively accumulated free radicals, such as reactive nitrogen species [24,45] and reactive oxygen species [24,30,45], that are generated in our body during physiological modulation of different organs and cellular activities can be destructive [24,30,45]. These species lethally attack the biological structures including genetic materials, proteins, lipids, and carbohydrates of localised body cells [45]. To prevent this oxidative damage, the consumption of kombucha and kefir is effective as they possess antioxidant activities, arising from the interdependent relations between the microbes and compounds within these beverages [24,32]. Polyphenols, which is a group of natural antioxidants in food, protect the body against oxidative stress by adjusting actions taken by enzymes and cell receptors [37,199]. Tea is renowned for its rich polyphenols, especially catechins in green tea and theaflavins and thearubigin in black tea [21,69], hence imparting antioxidant properties to kombucha. After fermentation, the antioxidative power of kombucha is improved as compared to tea, due to the microbial actions of the SCOBY. AAB and yeasts in kombucha release enzymes that catalyse the conversion of polyphenols in tea into smaller molecules of antioxidants [28], such as phenolic acids, phenolics, and flavonoids [21]. The antioxidant effects of polyphenols can be further boosted when they conjugate with GlcUA, an organic acid produced by the SCOBY, as the bioavailability and transport of polyphenols are enhanced [37]. On the other hand, kefir possesses different and wider ranges of polyphenols, mainly due to the usage of different fermentation substrates. For example, polyphenols found in soymilk kefir involve isoflavone analogues, tocopherols, saponins, chlorogenic acid, caffeic acid, and phenolic acid [29]. In milk kefir, the antioxidants comprise peptides (from milk casein hydrolysate) [32,35,200], amino acids (especially tyrosine, tryptophan, cysteine, and taurine), vitamins A and E [32,35], carotenoids, and enzymatic systems (superoxide dismutase, catalase, and glutathione peroxidase) [35]. Kefiran, a type of exopolysaccharide that is released by culture in kefir grains, has also been deduced to be one of the effective antioxidants in kefir [184]. Although being disparate in substrates, kombucha and kefir mutually share some types of antioxidative capabilities. For instance, the antioxidants in these beverages aid in preventing lipid peroxidation [30,32,36,123,201], ascorbate autoxidation [32], and protein oxidation [36,184].

The antioxidative power in kombucha varies with the starter cultures [21,22,24,25,147], metabolites [21,24,25,202], polyphenols [20,21,22,24,25,147,202], temperature [21], and fermentation time [20,21,22,24,25,147,202]. Kombucha inoculated with isolates of *S. cerevisiae* and AAB showed the highest scavenging activities against hydroxyl and DPPH radicals in black tea, but the performance in green tea was surpassed by a traditional kombucha liquid fermented by a native culture involving yeast (*Sd. ludwigii*, *S. cerevisiae*, *S. bisporus*, *Torulopsis* spp., and *Zygosaccharomyces* spp.) and other AAB [26]. The addition of wheatgrass juice as a composite ingredient to kombucha appears to further enhance its scavenging activity by >90% for DPPH and 12.8 μmol Trolox equivalents/mL for oxygen radical absorbance capacity [28]. Meanwhile, fermentation by kefir grains was found to increase antioxidant capacity (DPPH) of cow milk [30], soymilk [29], and goat milk [30,35]. An increment in DPPH from 0.63% to 3.24% in goat milk kefir was recorded by Yilmaz-Ersan et al. [35]. The milk kefir exhibited a stronger scavenging ability against free radicals than milk [35], likely due to the release of milk peptides from kefir grains [35]. The antioxidative capabilities of kefir were also maintained for at least 21 days during storage [32,34,35]. Güven et al. [31] suggested that the microbial activities of a kefir culture are highly possible to play a part in the antioxidant activities of kefir, which surpassed vitamin E, on the oxidative damage caused by carbon tetrachloride. Furthermore, Ozcan et al. [32] proposed that differences in antioxidative capabilities can result from different affinities of antioxidative compounds and varieties of kefir microbiota and metabolites. Like kombucha, the free-radical-scavenging performances of kefirs were significantly improved, when ingredients rich in antioxidants, i.e., steam-treated coffee pulp [34] and black and green tea [33], were added.

In general, fermentation enhances the antioxidant potential of beverages compared to the nonfermented counterparts. The antioxidants in kombucha and kefir protect endothelial cells by reducing oxidative stress, thus lowering the risks of atherosclerosis and heart attack [45]. Adequate antioxidants in the body can prevent these cardiovascular diseases (CVDs) by halting low-density lipoprotein (LDL) oxidation, regulating cholesterol metabolism, and aiding smooth muscle relaxation, hence, lowering blood pressure [37,203].

### 4.2. Reductions in Blood Pressure and Cholesterol

Associated with their antioxidant capabilities, patients who suffer from CVDs that are mainly caused by hypertension and hypercholesterolemia [40] benefit from the consumption of kombucha and kefir, which is believed to reduce blood pressure and cholesterol [43]. Kefir, especially the peptides, potentially reduces blood pressure through hindering actions of angiotensin I-converting enzyme (ACE). ACE causes hypertension as it breaks down bradykinin that encourages vasodilation and catalyses the formation of the angiotensin II hormone, leading to vasoconstriction and rising blood pressure [39]. The ACE inhibition was increased up to 82% and 93% by the bioactive peptides that were released by the respective *Lc. casei* and kombucha cultures of *Gb. oxydans* and *D. anomala* through enzymatic proteolysis in a 72 h milk fermentation [132]. Rats fed with kefiran produced by *Lb. kefiranofaciens* also had their ACE activities dropped notably in the serum to 19.8 units/L and thoracic aorta to 19.9 milliunits/mg of protein as compared with the control at 21.7 units/L and 23.2 milliunits/mg of protein, respectively [40]. In kombucha, theanine, the dominant amino acid in green tea, was found to relieve hypertension and reduce blood pressure in rats by Yokogoshi et al. [36,204].

Hypertension and the build-up of cholesterol can lead to blockages in blood vessels, increasing the risks of heart attack and stroke [47]. Fortunately, polyphenols in kombucha block pancreatic lipase in cholesterol and triacylglycerol absorption [37]. Catechins (polyphenols) reduce cholesterol levels by reducing solubility and encouraging faecal excretion of cholesterol and triglycerides [36]. Kombucha intake was found to reduce serum total cholesterol from 11.02 to 9.06 mmol/L and LDL from 0.48 to 0.32 mmol/L after 12 weeks in mice with a hypercholesterolaemic diet, possibly related to functional compound, D-saccharic acid-1,4-lactone (DSL) released by *Gluconacetobacter* spp. A4 [38]. Kefir consumption, in the respective forms of milk kefir [41], soymilk kefir [41], kefiran [40], and lyophilised *Lp. plantarum* powder that was extracted from Tibetan kefir grains [44] reduced cholesterol in the forms of total cholesterol, LDL cholesterol, and total triglyceride levels in rats [40,41,44]. In the studies of Liu et al. [41], Maeda et al. [40], and Wang et al. [44], cholesterol reduction was observed not only in the serum but also in other parts of the body (liver and thoracic aorta), indicating an overall decrease rather than redistribution in the body. On that account, both kombucha and kefir are highly potential candidates as antihyperlipidemic agents. The probiotic LAB play their roles by (a) preventing cholesterol absorption from the small intestine when they bind, incorporate, and assimilate cholesterol [41,44]; (b) stopping actions of cholesterol-synthesizing HMG-CoA reductase with some of their metabolites (such as propionic acid) [44]; and (c) deconjugating (bacterial bile salt hydrolase activity) bile acids that drain the cholesterol pool [41,44]. By releasing cholesterol-destroying enzymes, the cultures isolated from kefir grains assimilated cholesterol up to 62.5% and 84.2% during a 24 h incubation at 20 °C and a 48 h storage at 10 °C, respectively [42,43]. This behaviour of bacteria was proven to be strain-dependent, and some examples of the cholesterol-assimilating bacteria are *Lb. acidophilus*, *Lp. plantarum*, and *Lc. paracasei* and some *Bifidobacterium* strains [42].

### 4.3. Anti-Inflammatory and Modulating Immunity

Kefir and kombucha have been shown to possess anti-inflammatory properties [36,182]. For instance, kefir has been demonstrated to promote proliferation of cells that produce interleukin 10 (IL-10), a well-known anti-inflammatory cytokine [49]. Interestingly, the study of Vinderola et al. [49] suggested that this effect was more pronounced in the biogenics of kefir microbiota (peptides and polypeptides proteolysed from milk casein), rather than the kefir microbiota itself. Usually, nitric oxide (NO) is released by inducible nitric oxide synthase in macrophages as an immune response, but its accumulation is severely oxidative and inflammatory [45]. Research on oak kombucha has shown that it can provide sufficient levels of nitrite to control levels of inflammatory cytokines (TNF-alpha and IL-6) without damaging lipopolysaccharide [45]. This is due to the innate presence of antioxidative polyphenols [45], which can modulate gut microbiota and improve the intestinal barrier [205]. Guruvayoorappan and Kuttan [46] claimed that (+)-catechin played an essential role in enhancing the antioxidative and anti-inflammatory capabilities of kombucha, with respect to the suppression of lipopolysaccharide-activated macrophages from releasing NO by 62.2–74.3% and TNF-alpha by 18.5–78.6% for 5–25 µg/mL concentrations of (+)-catechin. Additionally, kombucha may alleviate inflammation-induced diseases, e.g., arthritis, rheumatism, and gout, as GIcUA production by the kombucha SCOBY is linked to the cartilage, collagen, and fluid that lubricates joints when the metabolite is converted into glucosamine, chondroitin-sulphate, and other acidic mucopolysaccharides and glucoproteins in the human body [37,47]. Milk kefir with immunosuppressive components may alleviate symptoms of intestinal inflammation in patients [39].

Apart from being immunosuppressive in cases of inflammation, the beneficial components in kombucha and kefir also modulate the immune system by boosting immunity. Vitamin C and B2, which are strong antioxidants found in kombucha, potentially aid and support the immune system [26,36,47,48]. Among rats that were orally administered with kefir, a specific intestinal mucosal immune response against cholera toxin was boosted, accompanied by a higher amount of total IgA cells at 281 mg/L from 206 mg/L and total IgG cells at 12,551 mg/L from 5578 mg/L in young adult rats [51]. The secretory IgA performs immune exclusion, blocking the entry of microbial pathogens and antigens by cooperating closely with the nonspecific defence mechanisms [49]. Similar findings were obtained in the study of Vinderola et al. [49], together with the increase in the number of IL-4+, IL-10+, and IL-6+ cells in the small intestine. For oral-administered kefir with a respective dilution ratio of 1/100 and 1/200, a higher amount of IgA+ cells at 99.5 and 86.7 N° IgA+ cells/10 fields and IgG+ cells at 38.6 and 35.6 N° IgG+ cells/10 fields, respectively, relative to the control with 81.2 N° IgA+ cells/10 fields and 35.0 N° IgG+ cells/10 fields, was also observed at the intestinal mucosal level of mice after two days [50]. Moreover, Vinderola et al. [49] emphasised that kefir intake increased the pathogen-ingesting activities of peritoneal and pulmonary macrophages, while kombucha, including glucuronic acid, lactic acid, acetic acid, and some antibiotic compounds, work collaboratively to regulate immunity and inhibit cancer proliferation [48].

### 4.4. Anticancer and Antimutagenic

Cancer prevention and improved immunity can potentially be achieved by consuming kombucha as it has antioxidant ability. The Russian Academy of Sciences and the Central Oncological Research Unit in Russia proposed that a daily intake of kombucha is correlated with strong protection against cancer [20] after conducting a population investigation [36]. Some functional components in kombucha and kefir have anticancer features, including polyphenols [52,53], gluconic acid [52], GlcUA [52], lactic acid [52], vitamins [42,52], genistein [42], peptides [56], and DSL. DSL inhibits glucuronidase, i.e., an enzyme indirectly related to cancers [52]. Kombucha also contains verbascoside, a phenylpropanoid glucoside that has anticancer properties [21] as it has prevented cell growth and induced apoptosis (programmed cell death) in both in vitro and in vivo models of human oral squamous cell carcinoma [54].

Given the association between cancer and mutations, the antimutagenic properties of kefir have garnered significant interest. Kefir extract was able to reduce mutagenicity of methyl methanesulfonate and sodium azide by 37% and 30%, more efficiently than milk by 12% and 9% and yogurt by 27% and 13%, respectively, potentially due to the higher content of antimutagenic components in milk fat, including isomers of conjugated linoleic, butyric, palmitic, palmitoleic, and oleic acids [55]. The antimutagenic-conjugated linoleic acids are helpful in reducing cholesterol content and inhibiting atherogenesis too [42]. The mutations induced by N-methyl-N’-nitro-N-nitrosoguanidine and 4-nitroquinoline-N’-oxide are also reduced with the presence of milk and soymilk kefirs, whereby the inhibition rates increased up to 62.0% and 89.3% for milk kefir and up to 45.7% and 68.8% for soymilk kefir, respectively [29].

Angiogenesis, the formation of new blood vessels from the pre-existing vascular bed, initiates the growth of cancer cells [27]. The treatment using kombucha for a human prostate cancer cell line (PC-3) was effective in reducing the cell viability, migration, and activities of angiogenic-stimulating molecules, such as HIF-1, VEGF, IL-8, and COX-2, in a dose-dependent manner [27]. For kombucha, the concentration ranged from 50 to 400 µg/mL, and both green tea and black tea kombucha had significant cytotoxicity, cell-killing ability, by inhibiting 50% cell growth on the Hep-2 (epidermoid carcinoma) cell lines with IC_50_ of 200 and 386 µg/mL, while green tea kombucha performed better by inhibiting A549 (lung cell carcinoma) cell lines too (IC_50_ = 250 µg/mL) [52]. The higher anticancer effects of green tea kombucha than black tea kombucha were described in the study of Cardoso et al. [21], and the cell growth of Caco-2 cells was more successfully inhibited by 50% for green tea kombucha, with lower values of GI_50_ at 40.93 µg/mL, as compared to 47.15 µg/mL for black tea kombucha [21], potentially due to the higher concentration of catechins and the presence of verbascoside [21]. The importance of biogenics was reported for kefir too. Both kefir and kefir cell-free fraction that were orally administered to mice with induced breast cancer slowed down tumour growth and raised the levels of IgA(+) cells [56]. A similar outcome was found by Liu et al. [57] when they used milk and soymilk kefir as anticancer agents in mice inoculated with sarcoma-180 tumour cells. Azoxymethane, a carcinogenic substance, harmed the mice with liver lesions and abnormal crypt formation in the colon in the studies of Sozmen et al. [58] and Cenesiz et al. [59]. The antioxidative capabilities of kefir could have contributed to these cases. It is believed that kombucha intake is a safe anticancer treatment because the cytotoxic and antiproliferative impacts of kombucha on noncancer cells were milder than on those of cancer cell lines [21].

### 4.5. Antimicrobial

The microbiota in both kombucha and kefir release organic acids as byproducts during the fermentation process. Organic acids are important in providing these beverages with their antimicrobial properties [36] as a low pH environment is unfavourable for pathogenic bacteria [108]. The antimicrobial potential of kombucha is claimed to be gifted by fermentation as unfermented tea or juice do not possess any antimicrobial potential [52,60,62]. Studies proved that kombucha inhibited *Bacillus cereus* [60,61,62], *Klebsiella pneumoniae* [62], *Enterococcus faecalis* [62], and *Listeria monocytogenes* [60]. Kombucha also comprises antifungal components as its performance in fighting against *C. albicans*, *C. krusei*, *C. glabrata*, *C. tropicalis* [153], *Aspergillus flavus*, and *Aspergillus niger* [60] was found to be brilliant [60]. *Escherichia coli* and *Staphylococcus aureus* have been deduced to be the most susceptible towards the antimicrobial actions of kombucha [60]. Deghrigue et al. [53] recorded the antimicrobial effects of black and green tea kombucha against Gram-negative bacteria, *E. coli* at 150 µg/mL, *Salmonella typhimurium* at 336 µg/mL, and *Pseudomonas aeruginosa* at 228 µg/mL. The green tea kombucha stood out with great impacts on Gram-positive bacteria, including *Micrococcus luteus* at 216 µg/mL, *S. aureus* at 280 µg/mL, and *S. epidermis* at 324 µg/mL [52]. The stronger antimicrobial capabilities of green tea kombucha were explained by the higher antibacterial catechins and acidity, plus the sole presence of antibacterial verbascoside in green tea [21]. In addition to organic acids, other metabolites, such as alkaloids, heterocyclic compounds, and esters or microbes, in kombucha also play important antimicrobial roles because neutralised kombucha still shows sufficient antimicrobial potential but poorer [60].

Kefir also contains organic acids together with various antimicrobial metabolites, including hydrogen peroxide, acetaldehyde, carbon dioxide, and bacteriocins that are released by bacteria present in kefir grains, particularly Lp. plantarum ST8KF [67]. *Lb. acidophilus* and *Lb. kefiranofaciens*, isolated from kefir in the study of Santos et al. [64], also showed great antimicrobial potential by inhibiting all the tested bacteria of *E. coli*, *L. monocytogenes*, *Salmonella typhimurium*, *Salmonella enteritidis*, *Shigella flexneri*, and *Yersinia enterocolitica*. Evidenced through scanning electron microscopy, the cell structures of Caco-2 cells (cultured enterocytes) were well protected from the attack of the *B. cereus* strain B10502 with the presence of kefiran [66], while in the study of Kakisu et al. [65], they proposed that the antimicrobial potential of kefiran against *B. cereus* was dose-dependent, because 1% kefiran was not effective enough to inhibit the pathogen, but 5% kefiran inhibited spore formation and decreased spore concentration. This is associated with the amount of organic acids released by the microbiota during fermentation, which is reflected by the changes in pH [65].

In short, all these antimicrobial potentials of kombucha and kefir denote that drinking these beverages may reduce or prevent a microbial attack within the body because their antimicrobial potentials were similar to some antibiotics, for example, gentamycin and ampicillin [63]. Kefir grains containing *Leuconostoc* spp., *Lb. delbrueckii* subsp. *lactis*, *Acetobacter* spp., *S. cerevisiae*, *Km. marxianus*, and *Km. lactis* were also studied to confirm the antimicrobial effects on diarrhoeal disease, urinary tract infection, *Salmonella*, and streptococcal and *Helicobacter pylori* infections [206]. These beverages can also be used as natural preservatives in other food products to replace the synthetic ones that may bring adverse health risks.

### 4.6. Antidiabetic

Diabetes is often associated with hyperglycaemia (high blood sugar) [68]. Type 2 diabetes, resulting from insufficient insulin action or secretion, is a widely recognised global health issue [72]. The state of being hyperglycaemic harms the body through exhibiting strong oxidative stress and long-term tissue damage and complications, e.g., liver–kidney dysfunctions [68]. Diabetic patients, lacking in insulin, may also face sudden weight loss due to glucose intake from the body through gluconeogenesis instead of assimilation in the small intestine, resulting in a decrease in and loss of muscle tissue and adipocytes [70]. It is also common to find serious inflammation in the bodies of diabetic patients [70].

The oral administration of kombucha to diabetic rats for 30 days led to lesser α-amylase enzymatic activity by 37% in the plasma and 52% in the pancreas with a glucose level that decreased by 50% when compared with the untreated ones [68]. Tanaka and Kouno [69] proposed that theaflavin in black tea kombucha blocks amylase activities in the digestive tract; therefore, the spike in postprandial glucose levels was also reduced. Catechin, which is inherited from green tea, performs better as an antidiabetic, together with the metabolites of kombucha, i.e., acetic acid and GlcUA [70]. With enough time and dosage, the capabilities of DSL, a metabolite produced in kombucha, in reducing blood sugar to almost the normal level and improving blood insulin from 7.12 to 13.43 microunits/mL in diabetic rats were demonstrated [71]. Hypoglycaemic effects were demonstrated by kombucha intake through the alteration of the gut microbiota. Being exclusively found in black tea kombucha after fermentation [21], the administration of pelargonidin-3-O-glucoside (Pg3G) at 150 mg/kg promoted glucose metabolism in diabetic mice by increasing the bacteroidetes/firmicutes ratio and boosting the number of short-chain fatty acid (SCFA)-producing bacteria [207], for example, *Lactobacillus*, *Bifidobacterium*, *Butyricicoccus*, and *Lachnospiraceae* _NK4A136_group [205]. The roles of SCFAs were demonstrated by the significant elevation in the abundance of *Prevotella* (SCFA producer) [205,207] and reinforcement of the intestinal barrier [205].

Kefir also possesses antidiabetic properties, as SCFAs, mainly acetic acid at 1.78–2.71 g/L and propionic acid at 0.57–0.59 g/L, were found within this functional beverage [208]. Induced by probiotics in kefir, including Lactobacillus and Bifidobacteria, gut microbes contributed to the release of insulinotropic polypeptides and glucagon-like peptide-l that boost the glucose uptake of muscle [73], thus reducing blood glucose. The consumption of kefir, containing *Sc. thermophiles*, *Lb. acidophilus*, *Lc. casei*, and *Bifidobacterium lactis*, for 8 weeks among diabetic patients significantly decreased insulin resistance that was evidenced by HOMA-IR, from 7.05 to 4.93 [72], while in another study using kefir with identical major microbiota, the sharp decrease in the parameters for diagnosing diabetes including fasting blood glucose, from 161.63 to 139.22 mg/dL, and glycated haemoglobin (HbA1C) of the patients, from 7.61 to 6.40, was significantly greater than in those patients who consumed conventional fermented milk, i.e., from 183.42 to 182.16 mg/dL and from 6.98 to 7.00, respectively [73].

### 4.7. Detoxification and Protection of Liver and Blood

Liver problems arising from diabetes, hepatotoxins, or carcinogens are chronic as liver is essential in the body as a physiological process modulator. Kabiri et al. [74] conducted a study to determine the effects of consuming kombucha on the protection of the liver by inducing liver damage in rats using thioacetamide, a toxin causing hepatic fibrosis. With kombucha treatment, each parameter indicating thioacetamide-induced liver damage, i.e., serum aspartate transaminase, alanine transaminase, alkaline phosphatase, and lactate dehydrogenase significantly dropped from 653, 767, 1594, and 1270 units/L to 131, 127, 947, and 808 units/L, respectively, possibly attributed to the abundant polyphenols (antioxidants) [37,74]. The innate production of GlcUA by *D. bruxellensis* and *G. intermedius* in kombucha contributes to its detoxifying capability through the elimination of drugs, bilirubin, and chemicals, as well as pollutants and excess steroid hormones through the glucuronidation process [75,76]. The detoxifying feature of the kombucha culture may also be on account of the adsorbent ability of the SCOBY, as proposed by Ismaiel et al. [77], who successfully reduced a toxin known as patulin up to 100% in an aqueous solution by using the SCOBY with *A. xylinum* as the dominant bacteria. Nonalcoholic fatty liver disease (NAFLD) is another liver disorder that is possibly protected with an adequate amount of these beneficial beverages to restore the liver by reducing inflammation [37] and modulating lipid metabolism [78]. Kefir at 140 mg/kg had markedly fewer activities of biochemical markers of hepatic injuries, especially for NAFLD, i.e., serum glutamate oxaloacetate transaminase and glutamate pyruvate transaminase, at 636.4 and 492.7 units/L, respectively, compared to the untreated ones at 1433.4 and 1183.8 units/L, and decreased levels of triglyceride by 26% and total cholesterol by 27% in the liver [78], which is also potentially related to the capabilities of kefir in reducing cholesterol content. The severity of liver lesions caused by the toxic azoxymethane was also decreased with kefir intake [58].

## 5. Future Perspectives of Kombucha and Kefir

The popularity of kombucha and kefir, along with the demand for their functional properties, has led to an increase in their production and, subsequently, the proliferation of starter cultures. The reusable and adaptive features of the cultures could serve as viable alternatives in various industries, including in the production of probiotic-rich foods and natural preservatives as well as packaging and waste management, potentially generating economic and sustainable benefits. Among the innovative options, one explorable approach is the use of diverse substrates. Comparing the substrates used in kombucha and kefir, it is evident that some substrates are unique to each of them. These unique substrates present an opportunity for further exploration into how they impact the microbial communities and chemical compositions of these drinks. While sugar initiates kefir fermentation, the exact process of kefir grain formation and the role of other substrate components remain unclear. Further elucidation of the metabolic pathways is therefore essential to understand the underlying mechanisms of fermentation and identify opportunities for process optimisation. Although the formation of SCOBY in kombucha is well understood, i.e., formed from cellulose produced by acetic acid bacteria and yeast, the exact nutrient source for optimal SCOBY growth has not been well defined. More research is needed to understand the SCOBY needs and its relationship with microbial diversity in kombucha. Despite the successful identification of some micro-organisms present in kombucha and kefir, there is still a considerable research gap in understanding how these microbial communities specifically contribute to health and the sensory characteristics of these drinks, particularly the impact on the overall drinking experience. As such, further research is necessary to better understand the interactions among substrates, microbial communities, and human physiology in order to optimise the health benefits and sensory qualities of these drinks.

## 6. Conclusions

Fermentation has brought about the creation of functional beverages, such as kombucha and kefir, which share a common feature of being produced by the action of starter cultures, namely SCOBY and kefir grains, respectively, in a sugar-containing liquid. In kombucha, the dominant micro-organisms are acetic acid bacteria and yeast, while in kefir, lactic acid bacteria and yeast play the dominant roles in the fermentation process. The microbial communities of these functional beverages are significantly different based on the substrates and origins of the starter culture used, resulting in overlapping yet distinct health benefits. Both beverages contain probiotics and polyphenols, which scavenge free radicals and protect the body from oxidative attacks, which may help prevent hypertension and atherosclerosis. Bioactive peptides, GIcUA, and catechins are some of the metabolites that may act as immunity modulators. Kombucha and kefir display anticancer properties, contributed by catechins and verbascoside in kombucha and antimutagenic components in kefir. These beverages may also protect the body from microbial attacks and have shown their potential in controlling diabetic conditions and liver problems. Addressing the challenges and limitations through ongoing research is crucial for recognising the potential benefits of kombucha and kefir. This could ultimately lead to advancements in the field, promoting longevity and well-being.

## Figures and Tables

**Figure 1 microorganisms-11-01344-f001:**
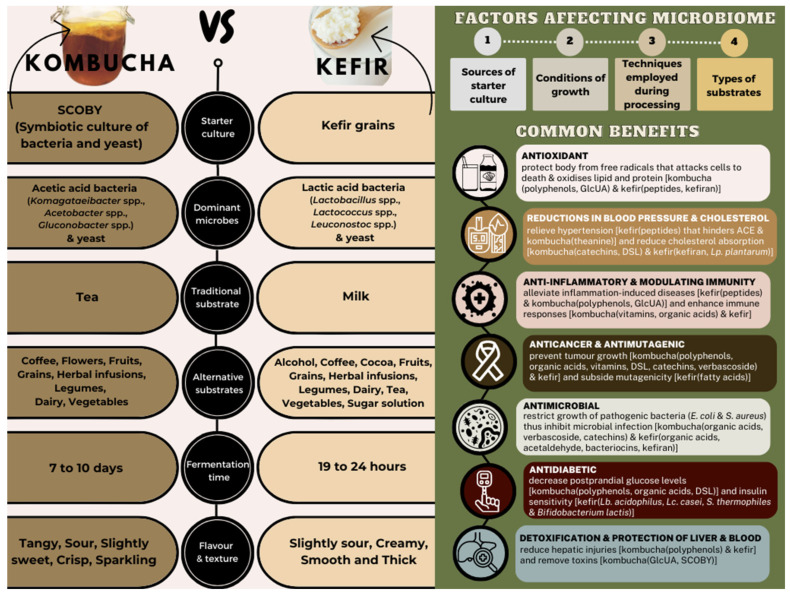
Graphical abstract summarising comparisons between kombucha and kefir fermentation, factors affecting microbiome, and common benefits.

**Figure 2 microorganisms-11-01344-f002:**
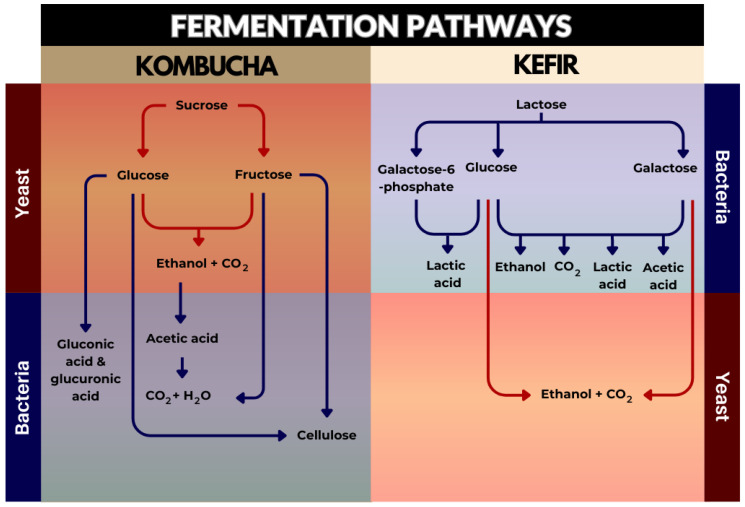
Summary of the main pathways involved in kombucha and kefir fermentation.

**Table 1 microorganisms-11-01344-t001:** Common benefits of kombucha and kefir.

Benefits	References	
	Kombucha	Kefir
Antioxidant	[20,21,22,23,24,25,26,27,28]	[29,30,31,32,33,34,35]
Reductions in blood pressure and cholesterol	[36,37,38]	[39,40,41,42,43,44]
Anti-inflammatory and modulating immunity	[26,36,37,45,46,47,48]	[39,49,50,51]
Anticancer and antimutagenic	[21,27,36,52,53,54]	[29,42,55,56,57,58,59]
Antimicrobial	[21,52,60,61,62,63]	[64,65,66,67]
Antidiabetic	[68,69,70,71]	[72,73]
Detoxification and protection of liver and blood	[37,74,75,76,77]	[58,78]

**Table 2 microorganisms-11-01344-t002:** Substrates used in kombucha and kefir production.

Substrates	References	
	Kombucha	Kefir
**Alcohol**		
Beer		[102]
**Coffee**		
Coffee	[103]	[34,104]
**Cocoa**		
Cocoa		[105]
**Flowers**		
Wax mallow	[6]	
**Fruits**		
Cactus pear	[106]	
Goji berry	[107]	
Coconut	[108]	[109]
Red grape	[62]	
Sour cherry	[110]	[89]
Apple	[111,112,113]	[89,114]
Snake fruit	[113]	
Pomegranate	[115]	[116]
Banana	[117]	
Melon		[100]
Strawberry		[100]
Papaya leaves	[118]	
Raisin		[114]
Fig		[114]
Pumpkin		[119]
Orange		[116]
**Grains**		
Rice	[20]	[120]
Barley	[20]	
**Herbal infusions**		
Lemon balm	[93]	
Thyme	[95]	
Lemon verbena	[95]	
Rosemary	[95]	
Fennel	[95]	[100]
Peppermint	[95]	
Stinging nettle	[121]	
Winter savory	[121,122]	
Yarrow	[96]	
Oak	[45]	
River red gum	[123]	
Mexican bay leaf	[123]	
Cinnamon	[61]	
Cardamom	[61]	
Shirazi thyme	[61]	
Ginger	[124]	
Nettles	[117]	
Willow		[125]
**Legumes**		
Soy whey	[126]	
Soy milk	[127]	[84,128]
**Dairy**		
Cheese whey	[129]	[130]
Milk (Unknown)	[99,131,132,133]	[16,89,134,135]
Milk (Cow)	[97,98,121,133,136]	[84,85,128,130,137,138,139,140,141]
Milk (Goat)		[140,141,142,143,144]
Milk (Sheep)		[142,143]
Milk (Camel)		[139]
Milk (Mare)		[142]
Milk (Bovine)		[145,146]
Milk (Caprine)		[146]
Milk (Ovine)		[146]
Milk (Buffalo)		[32]
**Tea**		
Black tea	[20,21,25,45,77,83,113,147,148,149,150,151,152,153,154,155,156,157,158,159,160]	[33]
Green tea	[21,25,148,152,155,156,158,160,161,162,163,164]	[33]
Oolong tea	[156,160]	
White tea	[162]	
Red tea	[162]	
Rooibos tea	[152]	
**Vegetables**		
Jerusalem artichoke	[165]	
Spinach	[166]	
Broccoli	[166]	
Wheatgrass	[28]	
Carrot		[100]
Onion		[100]
Tomato		[100]
**Sugar solution**		
Sucrose solution		[89,114]
**Others**		
Bee pollen	[167]	
Honey		[168]

## Data Availability

No new data were created or analysed in this study. Data sharing is not applicable to this article.

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
