# Peer review of "Fermented Beverage Benefits: A Comprehensive Review and Comparison of Kombucha and Kefir Microbiome"

_microorganisms, 2023, doi:10.3390/microorganisms11051344_

Round 1

Reviewer 1 Report

This is an interesting and well written work that shows the different microorganisms involved in the fermentation process of kombucha and kefir, as well as the potential health benefits of these products. My advice is to accept the manuscript after the authors address some details as described below:

L68-L69 - The word “enzymes” is repeated

It would be useful to summarize the main pathways of fermentation in kombucha and kefir with an image. 

Table 3 and 4 are well described, consider including some information from these works. I consider it might be useful for the authors, to enrich the manuscript.

-A systematic, complexity-reduction approach to dissect the kombucha tea microbiome. DOI: https://doi.org/10.7554/eLife.76401

-Metagenomic and phytochemical analyses of kefir water and its subchronic toxicity study in BALB/c mice. https://doi.org/10.1186/s12906-021-03358-3

Health benefits: This section is sufficiently described, however, some information regarding the effect of consuming kombucha and kefir on the gut microbiota should be included.

Reviewer 2 Report

Chong et al.'s review article "Fermented Beverages Benefits: A Comprehensive Review and Comparison of the Kombucha and Kefir Microbiome" is well written. This article will strengthen the scientific knowledge and constitute life easier for the large scientific community.  

Please simply improve English writing and style.

Reviewer 3 Report

  Manuscript Number: microorganisms-2347811, titled:

 Fermented Beverages Benefits: A Comprehensive Review and Comparison of Kombucha and Kefir Microbiome.

Review 1 – 8 April 2023

Dear Editor of Microorganisms

the manuscript is interesting but it has to be improved. The Authors have to be consistent with the guidelines of Microorganisms in the references section and use some recently published papers as a template. Some sub-section has to be improved. Some inaccuracy in the manuscript.

I suggest a major revision

To the Authors (in detail):

  1. the manuscript is interesting but it has to be improved. The Authors have to be consistent with the guidelines of Microorganisms in the references section and use some recently published papers as a template. Some sub-section has to be improved. Some inaccuracy in the manuscript.

  1. Introduction section, discuss briefly about other fermented beverages such as Boza and Cider [X1], kvass [X2]. Support this statement with proper references, please, find, read and discuss: [X1, X2]. List also other fermented bevereges and support your statement with proper references. After that, introduce your argument.

[X1] Screening and Molecular Identification of Lactic Acid BacteriaProducing β-Glucan in Boza and Cider.

Fermentation 2022, 8 (8), 350. https://doi.org/10.3390/fermentation8080350

[X2] Use of clarifying agents in technological process of kvass production

IOP Conference Series: Earth and Environmental ScienceOpen AccessVolume 613, Issue 123 December 2020 Article number 012100

DOI: 10.1088/1755-1315/613/1/012100

  1. Please, detail better the Kombucha and Kefir recipes and the production procedure;
  2. Line 330 and in the whole manuscript, when you indicate a temperature, separate the numeric value from the symbol: 10 °C and not 10°C;
  3. References section, ref 21, 23 and in the whole section, please, abbreviate the journal name with the proper abbreviation;
  4. Section 4 and sub-section, when you discuss about properties, please, include some numeric value (finding) of the works you have listed;
  5. References 41, 62 and in the whole manuscript, please, apply the correct binomial identification of the scientific names; the Genus and the species have to be italicized and the species in small letters;

  1. Please, write in blue color or evidence differently the corrections you will do.

I suggest a major revision

Regards.

  Manuscript Number: microorganisms-2347811, titled:

 Fermented Beverages Benefits: A Comprehensive Review and Comparison of Kombucha and Kefir Microbiome.

Review 1 – 8 April 2023

Dear Editor of Microorganisms

the manuscript is interesting but it has to be improved. The Authors have to be consistent with the guidelines of Microorganisms in the references section and use some recently published papers as a template. Some sub-section has to be improved. Some inaccuracy in the manuscript.

I suggest a major revision

To the Authors (in detail):

  1. the manuscript is interesting but it has to be improved. The Authors have to be consistent with the guidelines of Microorganisms in the references section and use some recently published papers as a template. Some sub-section has to be improved. Some inaccuracy in the manuscript.

  1. Introduction section, discuss briefly about other fermented beverages such as Boza and Cider [X1], kvass [X2]. Support this statement with proper references, please, find, read and discuss: [X1, X2]. List also other fermented bevereges and support your statement with proper references. After that, introduce your argument.

[X1] Screening and Molecular Identification of Lactic Acid BacteriaProducing β-Glucan in Boza and Cider.

Fermentation 2022, 8 (8), 350. https://doi.org/10.3390/fermentation8080350

[X2] Use of clarifying agents in technological process of kvass production

IOP Conference Series: Earth and Environmental ScienceOpen AccessVolume 613, Issue 123 December 2020 Article number 012100

DOI: 10.1088/1755-1315/613/1/012100

  1. Please, detail better the Kombucha and Kefir recipes and the production procedure;
  2. Line 330 and in the whole manuscript, when you indicate a temperature, separate the numeric value from the symbol: 10 °C and not 10°C;
  3. References section, ref 21, 23 and in the whole section, please, abbreviate the journal name with the proper abbreviation;
  4. Section 4 and sub-section, when you discuss about properties, please, include some numeric value (finding) of the works you have listed;
  5. References 41, 62 and in the whole manuscript, please, apply the correct binomial identification of the scientific names; the Genus and the species have to be italicized and the species in small letters;

  1. Please, write in blue color or evidence differently the corrections you will do.

I suggest a major revision

Regards.

Reviewer 4 Report

This is an overview on the fermentation of kombucha and kefir, comparing the production process of both of them, exploring in parallel the potential health benefits associated with the different starters involved in fermentation process. In general, the aim of this manuscript is clear. However, from my point of view, the usefulness and significance of this overview to justify the need for its publication is not very clear, since it just provides the existent knowledge. Furthermore, in such kind of reviews, a thorough discussion about the so far knowledge, as well as proposing some future prospects, would be very interesting and are totally missing from the present manuscript. Finally, English editing is strongly recommended, since there are several parts throughout the manuscript that are difficult to be followed, as well as many typo errors.

Other remarks

-L14-15. Please revise.

-L16-18. Not clear enough. Please revise.

-L43. Please revise.

-L56. Please revise.

-L66. Please revise.

-L94-96. Not clear. Please revise.

-L191-192. Please revise.

-L194. Please revise the term “microflora” to “microbiota” throughout the manuscript.

-L 246. “they attack and kill surrounding body cell” please revise.

-L245-267. A more rationale and thorough description is needed herein.

-L268-270. Please revise.

-L502. Please revise.

-L527-530. Please revise.

There are several parts throughout the manuscript that are difficult to be followed, as well as many typo errors.

Reviewer 5 Report

The review is interesting. In general, has a good structure. The tables and figures are well.

The references are complete and actualized, but the conclusion is very large. The authors should rewritten it

Round 2

Reviewer 3 Report

  Manuscript Number: microorganisms-2347811, titled:

 Fermented Beverages Benefits: A Comprehensive Review and Comparison of Kombucha and Kefir Microbiome.

Review 2 – 15 May 2023

Dear Editor of Microorganisms

the manuscript is interesting and the authors have included all my comments. The introduction well describes the state of the art.  M&M are well described and data are well commented. The bibliography is complete.

I suggest the publication in the current form

Regards.

Reviewer 4 Report

The revised manuscript has been strongly improved and is now suitable for publication. I have no further comments.

Reviewer 5 Report

The corrections were done